

# iTRAQ-based comparative proteome analyses of different growth stages revealing the regulatory role of reactive oxygen species in the fruiting body development of *Ophiocordyceps sinensis*

Xinxin Tong[*], Fang Wang[*], Han Zhang, Jing Bai, Qiang Dong, Pan Yue, Xinyi Jiang, Xinrui Li, Li Wang and Jinlin Guo

Key Laboratory of Standardization of Chinese Medicine, Ministry of Education; Key Laboratory of Systematic Research, Development and Utilization of Chinese Medicine Resources in Sichuan Province-Key Laboratory Breeding Base founded by Sichuan Province, Chengdu University of Traditional Chinese Medicine, Chengdu, Sichuan, China

[*] These authors contributed equally to this work.

Corresponding author
Jinlin Guo, guo596@cdutcm.edu.cn

## ABSTRACT

In this study, using an isobaric tags for relative and absolute quantitation (iTRAQ) approach coupled with LC-MS / MS and bioinformatics, the proteomes were analyzed for the crucial three stages covering the fruiting body development of *Ophiocordyceps sinensis*, including sclerotium (ST), primordium (PR) and mature fruiting body (MF), with a focus on fruiting body development-related proteins and the potential mechanisms of the development. A total of 1,875 proteins were identified. Principal Component Analysis (PCA) demonstrated that the protein patterns between PR and MF were more similar than ST. Differentially accumulated proteins (DAPs) analysis showed that there were 510, 173 and 514 DAPs in the comparisons of ST vs. PR, PR vs. MF and ST vs. MF, respectively. A total of 62 shared DAPs were identified and primarily enriched in proteins related to 'carbon transport and mechanism', 'the response to oxidative stress', 'antioxidative activity' and 'translation'. KEGG and GO databases showed that the DAPs were enriched in terms of 'primary metabolisms (amino acid/fatty acid/energy metabolism)', 'the response to oxidative stress' and 'peroxidase'. Furthermore, 34 DAPs involved in reactive oxygen species (ROS) metabolism were identified and clustered across the three stages using hierarchical clustering implemented in hCluster R package. It was suggested that their roles and the underlying mechanisms may be stage-specific. ROS may play a role in fungal pathogenicity in ST, the fruit-body initiation in PR, sexual reproduction and highland adaptation in MF. Crucial ROS-related proteins were identified, such as superoxide dismutase (SOD, T5A6F1), Nor-1 (T5AFX3), electron transport protein (T5AHD1), histidine phosphotransferase (HPt, T5A9Z5) and Glutathione peroxidase (T5A9V1). Besides, the accumulation of ROS at the three stages were assayed using 2,7-dichlorofuorescin diacetate (DCFH-DA) stanning. A much stronger ROS accumulation was detected at the stage MF, compared to the stages of PR and ST. Sections of ST and fruit-body part of MF were stained by DCFH-DA and observed under the fluorescencemicroscope, showing ROS was distributed within the conidiospore and ascus. Besides, SOD activity increased across the three stages, while CAT activity has a strong increasement in MF compared to the stages of ST and

PR. It was suggested that ROS may act in gradient-dependent manner to regulate the fruiting body development. The coding region sequences of six DAPs were analyzed at mRNA level by quantitative real-time PCR (qRT-PCR). The results support the result of DAPs analysis and the proteome sequencing data. Our findings offer the perspective of proteome to understand the biology of fruiting body development and highland adaptation in *O. sinensis*, which would inform the big industry of this valuable fungus.

## INTRODUCTION

*Ophiocordyceps sinensis* (Berk.), syn. *Cordyceps sinensis*, belongs to Ascomycetes and has been used as medicinal treatments and healthy food in some Asia countries over 2,000 years (*Lo et al., 2013*). It is commonly named as "Dong Chong Xia Chao", specifically parasitizing the larva of ghost moth caterpillars (*Thitarodes* spp.) and making a complex of the fungal stroma and the remains of the caterpillar (*Lo et al., 2013*; *Wei et al., 2020*). Over 20 bioactive ingredients have been reported, such as adenosine, cordyceps acid, ergosterol and polysaccharides, exerting multiple pharmacological effects, including anti-inflammatory, anti-tumor, immunomodulating and antioxidative activities, etc. (*Xu et al., 2016*). Recent years, overexploitation, habitat excavation and the upward of snow line aggravated the yield decreases of *O. sinensis*, resulting in extremely high price, approximately US $60,000 per kg for high quality products (*Li et al., 2019a*). The success of large-scale cultivation has only recently been achieved (*Li et al., 2019a*; *Shrestha, 2012*; *Zhang et al., 2018*). An understanding of the biology of the fruiting body development would enhance scientific research, impact the big industry and protect this precious resources for sustainable usage.

   *O. sinensis* infects the host larva. Then the larva progressively becomes stiff and coated with mycelia on the remaining exoskeleton of the insects (*Li et al., 2019a*), and a small stroma bud emerges from the head of the sclerotium and form the stalked fruiting body (*Li et al., 2019a*). During the infection, insect hosts often rapidly produce plenty of ROS to directly kill pathogens (*Shrestha, 2012*), as a response, pathogen fungi develop an adequate ROS antioxidant defense system (*Shrestha, 2012*). Previous study showed that oxidoreductase putatively involved in the ecdysteroid metabolism of insect molting may have a potential relationship with the fungal pathogenicity in *O. sinensis* (*Xia et al., 2017*). Besides, the sclerotial differentiation state in *Sclerotium rolfsii* concurred with increasing levels of lipid peroxides (*Georgiou et al., 2006*). In *Morchella importuna*, the MAPK signaling pathway was activated and passed the signal from an area of high oxidative stress to a low area to initiate sclerotial formation along with the increasing levels of SOD (*Liu et al., 2018*). Recently, the morphologic studies showed that the numbers of hyphal bodies and the conversion of hyphal bodies into hyphae may play important roles in the mummification of the injected larvae (*Guiqing, Richou & Li, 2019*; *Li et al., 2020*).

*O. sinensis* naturally inhabits the alpine environments of the Qinghai-Tibetan Plateau with an average altitude of over 4,000 m (*Qin et al., 2018*). The ecological conditions have been demonstrated to initiate and promote the formation of the fruiting body (*Pöggeler, Nowrousian & Kück, 2006*). Both orientation and position of the neck on perithecium were light-dependent in *Neurospora crassa* (*Oda & Hasunuma, 1997*), and the number of protoperithecia greatly increases under blue-light illumination (*Degli Innocenti & Russo, 1983*; *Busch et al., 2003*; *Lara-Ortíz, Riveros-Rosas & Aguirre, 2003*). Light controls the balance of asexual versus sexual reproduction of *Aspergillus nidulans* (*Lara-Ortíz, Riveros-Rosas & Aguirre, 2003*). Omics studies promoted to understand the biology of the fungus guided by the ecological stimuli (*Li et al., 2019a*). Genome analysis of the highland adaption showed that the signals of positive selection for genes encoding for peroxidases in *O. sinensis* compared to other plain-dwelling fungi, probably contributing to the detoxification of strong ROS induced by high intensity UV (*Xia et al., 2017*). NADPH oxidase-generated ROS regulates sexual development in *A. nidulans* (*Lara-Ortíz, Riveros-Rosas & Aguirre, 2003*). The change in the distribution of ROS caused by the *sod-1* mutation was found to be an important factor to cause loss of the light-induced perithecial polarity (*Yoshida & Hasunuma, 2004*; *Belozerskaya et al., 2012*), suggesting that intracellular ROS may function as a novel light signal transducer in the perithecial polarity. Besides, the 12 shared DEGs (differentially expressed genes) detected among all five comparisons of adjacent growth stages of *O. sinensis* were primarily enriched in the terms of 'the response to oxidative stress' and 'peroxidase activity' (*Li et al., 2019b*). Of the 18 candidates associated with the fruiting body development, most were involved in terms of 'oxidative stress' and / or 'osmotic response' in *A. nidulans* (*Krijgsheld et al., 2013*). Our previous study showed that some genes involved in ROS system, such as SOD, NADPH oxidase (NOX) and cytochrome oxidase, might play important roles in light-dependent morphogenesis on the perithecia, ascus or perithecia formation (*Tong et al., 2020*). Based on above evidences, ROS may serve as an active regulator of the fruiting body formation and differentiation in *O. sinensis* guided by environmental stresses.

In this study, *O. sinensis* samples of three major developmental stages were harvested from the artificial cultivation workshop and comparative proteomic analyses was conducted using iTRAQ technology (*Gan et al., 2007*). In particular, we focused on the differential expression of fruiting body development related proteins and the expression profiles of ROS-related proteins in *O. sinensis*. This study provides an important insight into the crucial development-related proteins and the potential regulatory mechanism of ROS metabolism system in fungal pathogenicity and fruiting body development, which would further benefit the large-scale cultivation and the sustainable development of this precious bio-resource.

## METHODS

### Collection of *O. sinensis* samples in three major growth stages

Three major developmental stages of *O. sinensis* were harvested from the artificial cultivation workshop at our lab in Chengdu University of Traditional Chinese Medicine

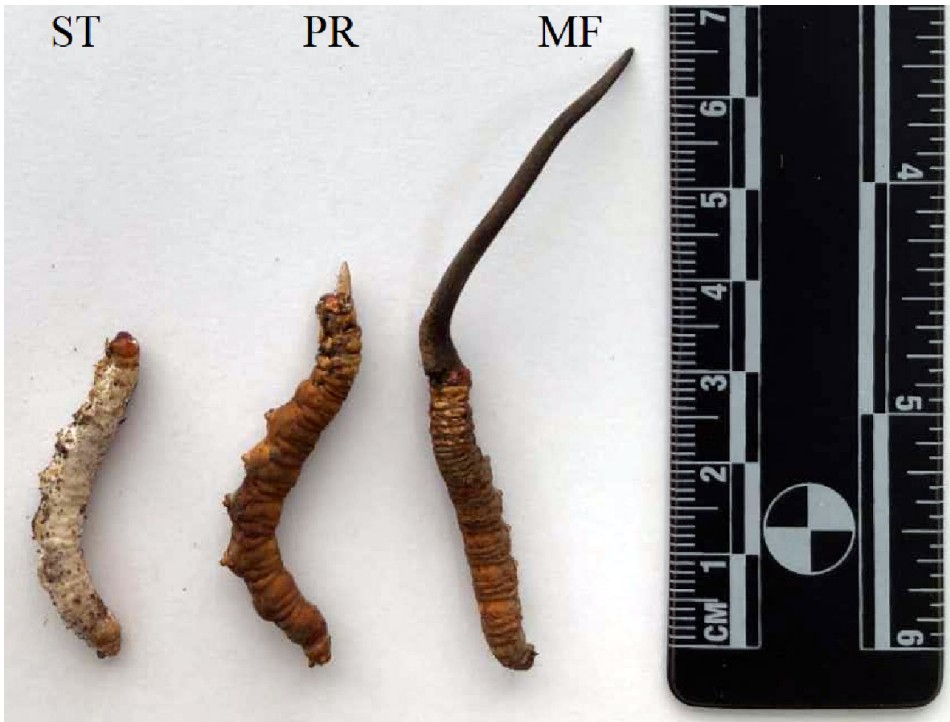

**Figure 1** ***O. sinensis* samples collection for proteome sequencing.** Different developmental stages of *O. sinensis*. ST, sclerotium (mummified larva) before stroma development; PR, sclerotium with initial stroma (stroma < 1 cm); MF, fruiting body with mature perithecia, ascus and ascospores.

(TCM) (Fig. 1). The mummified larvae coated with mycelia before stroma development were designated as the sclerotium (ST). The samples of stroma with lengths <1 cm were designated as the primordium (PR), and the fruiting body with mature ascus and ascospores was designated as the mature fruiting body (MF).

## Protein preparation

Each sample (0.1 g) was ground into powder in liquid nitrogen, lysed in a solution with 100:1 (w/v) urea lysis buffer containing 8M Urea, 2 mM EDTA, 10 mM iodoacetamide (DTT), 25 mM iodoacetamide (IAA), 1% protease inhibitor cocktail (Minipore), and sonicated for 1 min (2 s, 4 s). The proteins in the pooled supernatant was precipitated by addition of 3 times volume of acetone and incubated at −20 °C for 2 h. Following centrifugation at 12,000 rpm for 15 min at 4 °C, the pellet was collected and then air dried. Protein was extracted by resuspending the dry pellet in UT buffer (8 M Urea, 100 mM Triethyl ammonium Bicarbonate, TEAB). Protein concentration was determined with a Modified Bradford Protein Assay Kit (Bio-Rad, Hercules, CA, USA) according to the manufacturer's instructions, and then samples were kept in −80 °C prior to use.

## Peptide isobaric labeling

For each sample, 100 μg protein was used for digestion. The protein solution was reduced with 10 mM DTT for 1 h at 37 °C and alkylated with 55 mM iodoacetamide (IAM) for

30 min at room temperature in darkness. For trypsin digestion, the protein sample was diluted by adding 100 mM TEAB to obtain urea concentrations less than 2 M. Then the protein suspensions were digested with Sequencing Grade Modified Trypsin (Promega, Madison, WI, USA), 1/50 protein mass for the first digestion at 37 °C overnight, and then with trypsin (1/100 protein mass) for a second digestion for 4 h.

Following trypsin digestion, protein samples were desalted using Strata X SPE column (*Phenomenex Inc*; Torrance, CA, USA) and vacuum-dried using a SpeedVac concentrator (Thermo, San Jose, CA). Peptide was reconstituted in 20 μL 500 mM TEAB and processed according to the manufacturer's protocol for 8-plex iTRAQ kit (Sigma, Aldrich, USA). Briefly, one unit of iTRAQ reagent was all added to peptide solution after thawed and dissolved in 50 μL isopropanol. The peptide samples were incubated with 100 μg for 2 h at room temperature and pooled, desalted and dried by vacuum centrifugation.

## HPLC fractionation and high-resolution LC-MS/MS analysis based on Q exactive

The dried and labeled peptide was reconstituted with HPLC solution A (2% acetonitrile, ACN), pH 10, and then fractionated by high pH reverse-phase HPLC using Waters Bridge Peptide BEH C18 (130 Å, 3.5 μm, 4.6 × 250 mm). The peptides were eluted with a liner gradient of 2–98% ACN in pH 10 at a speed of 0.5 ml/min over 88 min into 60 fractions, and then the peptides were condensed into 20 fractions and dried by vacuum centrifugation. The peptide fractions were desalted using Ziptip C18 following the manufacturer's instructions, and finally dried under vacuum. The samples were stored at −20 °C till MS analysis was performed.

The iTRAQ-labeled samples were reconstituted in 0.1% formamide (FA), directly loaded into an Acclaim PepMap® 100 C18 reversed-phase pre-column (3 μm, 100Å, 75 μm × 2cm, Thermo Fisher Scientific, San Joes, USA ) at 5 μL/min in 100% solvent A (0.1 M acetic acid in water), and analyzed by NanoLC 1000 LC-MS/MS using a Proxeon EASY-nLC 1000 coupled to Thermo Fisher Q Exactive (Thermo Fisher Scientific, San Joes, USA). Peptide sequential separation using a reversed-phase pre-column a (Acclaim PepMap® RSLC 139 C18, 2 μm, 100Å, 50 μm × 15 cm) was conducted using a gradient of an increase from 0% to 8% solvent B (0.1% FA in 98% ACN) over 5 min, 8% to 25% solvent B over 35 min, 25% to 98% solvent B during 10 min, 5 μL / min in 100% solvent A (0.1 M acetic acid in water) and kept in 98% in 8 min at a constant flow rate of 300 nl/min on an EASY-nLC 1000 system. The eluent was sprayed with an NSI source at an electrospray voltage of 2.5 kV. The mass spectrometer was operated in data-dependent mode to switch automatically between MS and MS/MS. Full-scan MS spectra (from m/z 350 to 1,800) were acquired in the Orbitrap with a resolution of 70, 000. Ion fragments were detected in the Orbitrap at a resolution of 17, 500. MS data were obtained by selecting the 15-most-abundant precursors ions present in the survey scan (300–1,800 m/z) for decision-tree-based ion trap higher-energy collisional dissociation (HCD) fragmentation. The collision energy was set at 32% for HCD. Dynamic exclusion duration was 10.0 s.

## Data processing

The MS/MS raw data were analyzed by using the Sequest software integration in Proteome Discoverer (version 1.3, Thermo Scientific) and searched against *Ophiocordyceps sinensis* (strain Co18 / CGMCC 3.14243). The parameters were as follows: trypsin chosen as the cleavage enzyme, two missed cleavages, carbamidomethyl set as a fixed modification; and oxidation (M), N-Term Acetylation, as well as iTRAQ labeling were designated as variable modifications. The searches were performed using a peptide mass tolerance of 20 ppm and a product ion tolerance of 0.05 Da, resulting in 5% false discovery rate (FDR).

For visualization, PCA was performed. For the determination of DAPs, fold change was calculated as the average values of four biological repeats. Subsequently, proteins with a change ratio >1.2 and *p*-value <0.05, were considered to be significantly changed.

To determine the functional property of the proteins, their sequences were mapped to Gene Ontology (GO) annotation using UniProt-GOA database (http://www.ebi.ac.uk/GOA/) (*Camon et al., 2004*). Kyoto Encyclopedia of Genes and Genomes (KEGG) database (http://www.genome.jp/kegg/) and the COG (Cluster of Orthologous Groups) database (http://www.ncbi.nlm.nih.gov/COG/) were used to classify and group the proteins (*Gerlich & Neumann, 2000*). Firstly, KEGG online service tools KAAS was used to annotate protein's KEGG database description. Then mapping the annotation result on the KEGG pathway database using KEGG online service tools KEGG mapper (http://geneontology.org/). For each category, a two-tailed Fisher's exact test was employed to test the enrichment of the DAPs against all identified proteins. Correction for multiple hypothesis testing was carried out using standard false discovery rate control methods and with a corrected *p-value* <0.05 is considered significant. The expression patterns of DAPs involved in ROS-related GO annotation with *p-value* <0.05 across the growth stages were further clustered by one-way hierarchical clustering implemented in hCluster R package (Euclidean distance, average linkage clustering). Cluster membership were visualized by a heat map using the "heatmap" function from the R-package.

## Reactive oxygen species (ROS) measurement

To detect ROS level in the samples derived from the three stages, DCFH-DA (Nanjing Jiancheng Bioengineering Institute, Nanjing, China) for reactive species staining was performed according to the manufacturer's instructions. In brief, 0.5 g tissue was added with 2 ml Phosphate Buffered Saline (PBS) buffer containing 137 mM NaCl, 2.7 mM KCl, 1.5 mM $KH_2PO_4$, 8 mM $Na_2HPO_4$ and fully homogenized. Cell suspension was collected and centrifuged at $500 \times$ g for 10 min. Then cell pellets were collected and washed twice with cooled PBS and resuspended in 1ml PBS at a density of at least $1 \times 10^6$ cells per EP tube. Cell suspension was incubated with 2 μl 10 mM DCFH-DA for 30 min at 37 °C. Finally, the fluorescence intensity was detected by using a Flex station 3 (Molecular Devices, Sunnyvale, CA, United States). The fluorescence was excited at 488 nm, detected at 525 nm.

Intracellular production of ROS was detected by DCFH-DA (Nanjing Jiancheng Bioengineering Institute, Nanjing, China) according to the manufacture's instruction. Slices of tissues from the samples of ST and fruiting body part of MF were treated with

5 µM DCFH-DA in a 50 mM sodium phosphate buffer (pH 7.4, PBS) and incubated for 20 min at 37 °C. This non polar compound is actively taken up by cells and converted by esterases in DCFH, a non-fluorescent molecule, which is rapidly oxidized to the highly fluorescent DCF by intracellular peroxides. The samples were observed under a Leica DM6B fluorescence inverted microscopy at 488 nm excitation and photographed.

## SOD and CAT activities assay

SOD activity was evaluated by using SOD activity assay kit (Nanjing Jiancheng Bioengineering Institute, Nanjing, China) according to manufacture's instruction, based on the auto-oxidation of hydroxylamine, with the developed blue color then measured at 560 nm. Results are expressed as units of SOD/mg protein and calculated based on the formula: SOD activity (U/mg) $= (A_{light} - A_{measure})/50\% \times A_{light} \times V_{tissue} \times$ tissue mass (mg).

CAT activity was detected by using CAT activity assay kit (Nanjing Jiancheng Bioengineering Institute, Nanjing, China) according to manufacture's instruction. The decomposition of $H_2O_2$ by CAT was stopped by the addition of ammonium molybdate. The remaining $H_2O_2$ was then reacted with ammonium molybdate to generate a pale-yellow complex, which was measured at 405 nm (*Wei et al., 2018*). The CAT activity was calculated based on the formula: CAT activity (U/mg) $= [(Aself\text{-}control - A\ measure)/Ablank] \times 650/$protein concentration(mg / ml). The protein concentration of each sample was determined using BCA protein assay kit (Thermo Fisher Scientific, Waltham, MA, USA).

## qPCR validation

Total RNA was extracted from snap-frozen samples using TRIzol reagent (Tiangen Biotech, Beijing, China), and the RNA was reverse-transcribed using the FastKing RT Kit with gDNase (Tiangen Biotech, Beijing, China). The qRT-PCR was performed by using Ultra SYBR qPCR Mixture (Cwbiotech, Co., LTD, Beijing, China) on Bio-Rad CFX 96 Real-Time PCR Detection System ( Hercules, CA, USA). Each 20 µL reaction system contained 2 µL cDNA (50 ng/µL), 1 µL forward primer (10 µmol/L), 1 µL reverse primer (10 µmol/L), 10 µL 2 × Ultra SYBR qPCR Mixture. Thermal cycling conditions were pre-incubation at 95 °C for 30 s, followed by 35 cycles of denaturation at 95 °C for 5 s, annealing at 55 °C for 30 s, and an extension at 95 °C for 15 s. 18s rRNA (GenBank accession number: FM164742.1) was used as the internal reference gene to normalize the gene expression data. Six DAPs were randomly selected to evaluate the consistency of the mRNA expression level. The gene-specific primers for qPCR were designed by Primer Premier 5.0 software (Premier Biosoft International, CA, USA) and listed in Table S7. The relative expression levels were calculated by using the $2^{-\Delta\Delta Ct}$ method . Three biological replicates were set for each treatment.

## Statistical analysis

Proteome analyses were carried out for four independent biological repeats for each sample. Other experiments were performed using three independent biological repeats and each biological repeat was performed at least three technical repeats. All results are presented as the mean ± SD. Two-tailed Student's *t*-tests and graphs were performed using GraphPad

**Table 1  A number of spectra, peptide and protein identified by iTRAQ.**

|      | Total spectra | PSM   | Unique spectra | Peptide | Unique peptide | Protein group |
|------|---------------|-------|----------------|---------|----------------|---------------|
| ST   | 470465        | 60770 | 57731          | 25954   | 25114          | 1324          |
| PR   | 472920        | 46426 | 42031          | 18563   | 17887          | 1257          |
| MF   | 518917        | 46426 | 42031          | 20660   | 19904          | 1275          |

Notes.
ST, The mummified larvae coated with mycelia before stroma development; PR, the samples of stroma with lengths <1 cm; MF, the fruiting body with mature ascus and ascospores.

Prism v 6.0c software (GraphPad Software, Inc.). A value of *P-value* <0.05 was considered to be statistically significant.

## RESULTS AND DISCUSSION

### General information on iTRAQ analysis

To explore the patterns of proteome during the development, comparative proteome analysis was performed from three growth stages of artificially cultivated *O. sinensis* (Fig. 1). A total of 1,875 proteins were identified using iTRAQ in these samples of different stages. We detected 42,031, 57,731, 49,886 unique spectra, 18,563, 25,954, 20,660 peptides and 17,887, 25,114, 19,904 unique peptides in ST, PR and MF, respectively (Table 1). In this study, the most spectra results had a mass error within ± 5 ppm, indicating that the mass spectrometer's mass accuracy was normal (Fig. S1).The length of the peptides identified were distributed between 7 aa and 12 aa in (Fig. S2) and 90% of the peptide length was distributed within 24aa. MS data has been deposited in iProX (Integrated Proteome Resources, http://www.iprox.org/) with the primary accession code PXD021260.

### Proteome profile of the three stages of *O. sinensis*

*O. sinensis* has a complex life cycle. The caterpillar infection by the fungus, primordium induction and fruiting body development remain difficult task, due to the fact that the molecular basis of its lifestyle remain cryptic (*Li et al., 2019a*). To investigate the protein profiles during the growth, samples from three crucial growth stages including ST, PR and MF, were submitted for proteome analysis. Principal component analysis (PCA) revealed that the stages of PR and MF were grouped together, while the stage of ST remained separate, indicating that the differentiated stages PR and MF have much more similar protein expression profiles than the undifferentiated ST stage, similar to the result of transcriptome data (Fig. S3) (*Li et al., 2019b*; *Tong et al., 2020*). Based on a fold change >1.2 and a *p-value* <0.05, 1343 significant DAPs were identified (Table S1). There are 510, 173 and 514 DAPs detected in the comparisons of ST vs. PR, PR vs. MF and ST vs. MF, respectively (Table S2). 62 shared DAPs were identified among the three different growth stages (Table S3) and are primarily included in terms of 'carbon transport and mechanism', 'the response to oxidative stress', 'antioxidative activity', 'translation' and protein/amino acid synthesis. For examples, AhpC/TSA family protein (T5A9T9) and flavin-containing monooxygenase (T5AHW7) were oxidoreductases, playing roles in the response to oxidative stress. Phosphotransferase (T5AFE3) functions in carbohydrate transport, which is responsible for nutrients up-take to the cells of the developing fruiting

body (*Zhong et al., 2018*). Eukaryotic translation initiation factors (T5ADW2, T5AAF6 ) are relevant with protein synthesis. The mutant of eukaryotic translation initiation factor EF-1 α resulted in the lack of ascospores or perithecia in *Podospora anserine* (*Silar et al., 2001*). Besides, some uncharacterized proteins were also identified. Overall, the shared DAPs analyses indicated that the primary metabolisms and ROS system may play important roles in the developmental process. The roles of ROS and its underlying mechanisms needed to be further studied.

DAPs analysis showed that there are 419 up- and 91 down- regulated DAPs in the ST vs. PR comparison , 98 up- and 75 down-regulated DAPs in PR vs. MF, 441 up- and 73 down-regulated DAPs in ST vs. MF (Fig. 2, Table S2). Based on both KEGG and GO databases, we analyzed the enrichment of DAPs of the three growth stages (Fig. 3, Table S4 and Table S5). In ST vs. PR, DAPs were mainly enriched in GO terms of 'structure molecule activity' (GO:0005198) in cellular component, 'mitochondrial protein complex' (GO:0098798), 'nucleosome' (GO:0000786) and 'DNA-protein complex' (GO:0032993) in molecular function, 'amide / peptide transport'(GO:0042886, GO:00015833), 'translation' (GO:0006412), 'cellular amide metabolic process' (GO:0043604 ) in biological process (Fig. 3A, Table S4), indicating that various proteins could be synthesized during the infection. In PR vs. MF, DAPs were primarily enriched in terms of 'antioxidant activity' (GO:0016209) in MF, 'DNA packing complex' (GO:0044815), 'protein-DNA complex' (GO:0032993) and 'nucleosome' (GO:0000786) in cellular component, 'cellular oxidant detoxification' (GO:0098869), 'the response to oxidant stress/chemical stress/stress' (GO:0006979, GO:0042221, GO:0006950), 'the regulation of protein localization' (GO:0032880), 'amide/peptide transport' (GO:0042886, GO:0015833) in biological process, indicating amino / energy and ROS metabolism might play critical roles in the fruiting body development guided by environmental stimuli (Fig. 3B, Table S4). In ST vs. MF, DAPs were mainly enriched in terms of 'inner mitochondrial membrane protein complex'(GO:0098800), 'extracellular region' (GO:0005576), 'mitochondrial protein complex' (GO:0098798) in cellular component, 'peptide / amide biosynthetic process'(GO:0044455, GO:0043604), 'translation'(GO:0006412), 'generation of precursor metabolites and energy' (GO:0006091) in biological process, indicating an increased demand of energy, new proteins and metabolites in fruiting body development (Fig. 3C, Table S4). In particular, ROS metabolism system would affect the development, but the underlying mechanism remains to be clarified.

To further understand biological functions of the DAPs, pathway-based analyses was performed. KEGG pathway enrichment analysis showed that aminoacyl-tRNA biosynthesis, ribosome and proteasome showed significant enrichment in the comparison of ST and PR (Fig. 4A, Table S5), among which the pathways of 'proteasome'(map03050), 'glycan degradation' (map00511), 'aminoacyl-tRNA' (map00970) were up-regulated in ST compared to PR. Aminoacyl-tRNAs are the active substrates for protein synthesis. The pathway of 'aminoacyl-tRNAs' was up-regulated in ST, indicating protein synthesis would be more active during the host-pathogen interaction (*Tong et al., 2020*). The ubiquitin/26S *proteasome* system was implicated in plant-*pathogen* interaction (*Dielen et al., 2010*). In *O. sinensis*, approximately 35% of proteases detected have a signal peptide, indicating

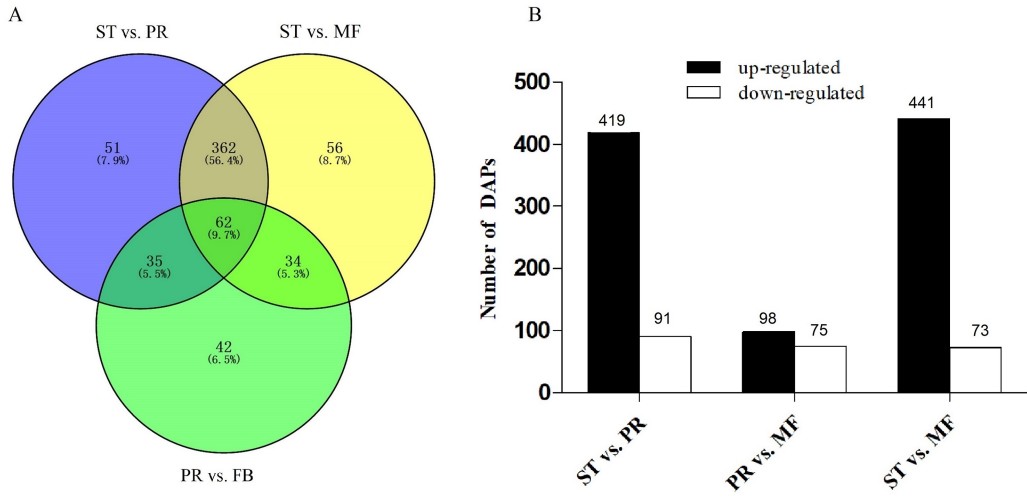

**Figure 2** **Analysis of differentially accumulated proteins (DAPs) between different growth stages in _O. sinensis_.** (A) Venn diagram of DAPs comparing between different growth stages from _O. sinensis_. ST, sclerotium (mummified larva) before stroma development; PR, sclerotium with initial stroma (stroma < 1 cm); MF, fruiting body with mature perithecia, ascus and ascospores. (B) The number of DAPs is shown on the top of histograms. statistics of DAPs from _O. sinensis_ between different growth stages.

a more efficient usage of secreted proteases in this fungi to adapt to the specific insect cuticles (_Xia et al., 2017_). Here, some proteases were detected to be highly up-regulated in ST compared to PR (Table S6), such as serine protease (T4ZXQ6), 26S proteasome regulatory subunit (T5AL30, T5AFG8), proteasome regulatory beta unite (T5AA86, T5A8T1) and proteasome regulatory alpha unite (T5A611). Serine protease in sclerotium may be beneficial for fungus to infect its host through digesting protein component of insect cuticle (_Zhao et al., 2005_). Two cuticle-degrading serine proteases have been obtained in _O. sinensis_ (_Zhang, Liu & Wang, 2008_). These results showed that proteases would play important roles in pathogen-host interactions in this fungus. Few related reports has been reported yet. In PR vs MF, the DAPs were enriched in pathways of 'phenylalanine, tyrosine and tryptophan biosynthesis'(map00400, map00350), 'protein processing in endoplasmic reticulum (ER)' (map04141), 'peroxisome(map04146)and fatty acid degradation' (map00071), among which the pathways of 'phenylalanine', 'tyrosine and tryptophan biosynthesis' and 'peroxisome' were up-regulated in MF compared to that in PR (Fig. 4B, Table S5), indicating that a variety of proteins may be bio-synthesized and needed for the fruit-body maturation (_Li et al., 2019b_; _Tong et al., 2020_). For examples, phospho-2-dehydro-3-deoxyheptonate aldolase (T5ABF2) remarkably increased in MF compared to PR. This protein is the first critical step for the shikimate pathway, which is involved in the biosynthesis of aromatic amino acids like phenylalanine and tyrosine. Mutants of the corresponding genes resulted in significantly decreases of fruit-body formation and ascosporogenesis in _A. nidulans_ (_Krappmann & Braus, 2003_), suggesting that shikimate pathway may participate into controlling fruit-body maturation in _O. sinensis_. Besides, the mutant of tryptophan synthase-encoding gene _trpB_ or the histidine biosynthesis gene

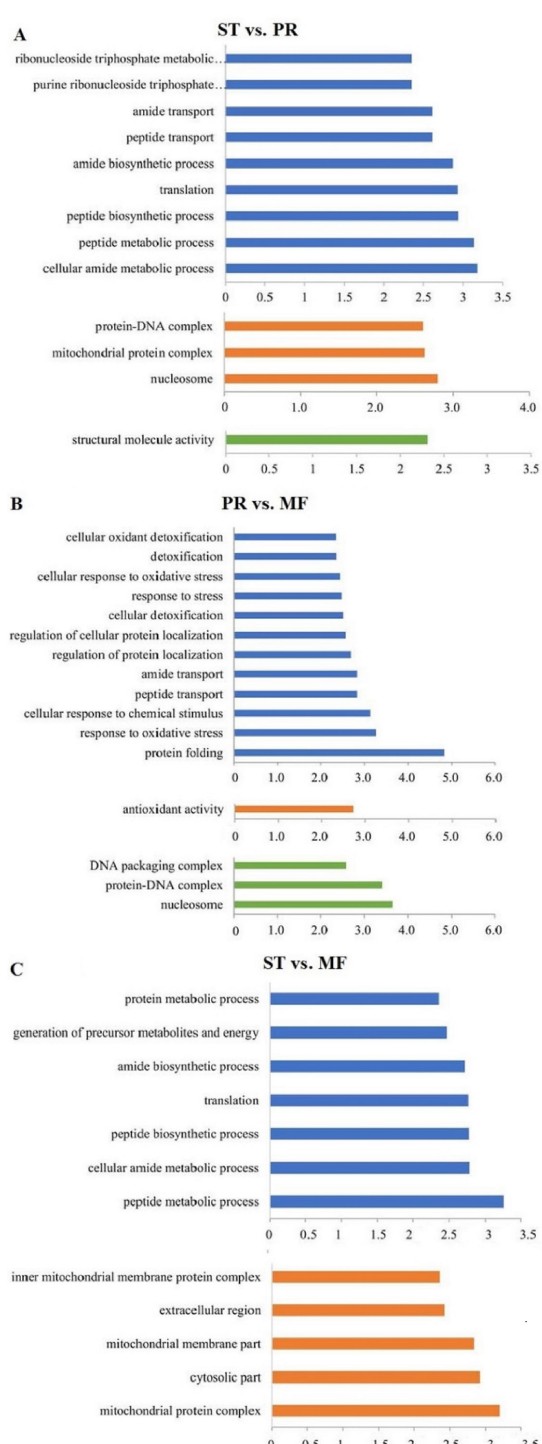

**Figure 3** **The most enriched GO functional classification of DAPs between different growth stages in *O. sinensis*.** The most enriched GO functional classification of DAPs in the ST vs. PR comparison (A) , in the PR vs. MF comparison (B) and in the ST vs. MF comparison (C). Only the significant GO terms (p < 0:005) were shown. *X*-axes represent the enrichment score (−Log10 *P-value*)) of top GO terms enriched among DAPs. The blue bars represent biological processes; the orange bars represent cellular components; the green bars represent molecular functions. ST, sclerotium (mummified larva) before stroma development; PR, sclerotium with initial stroma (stroma < 1 cm); MF, fruiting body with mature perithecia, ascus and ascospores.

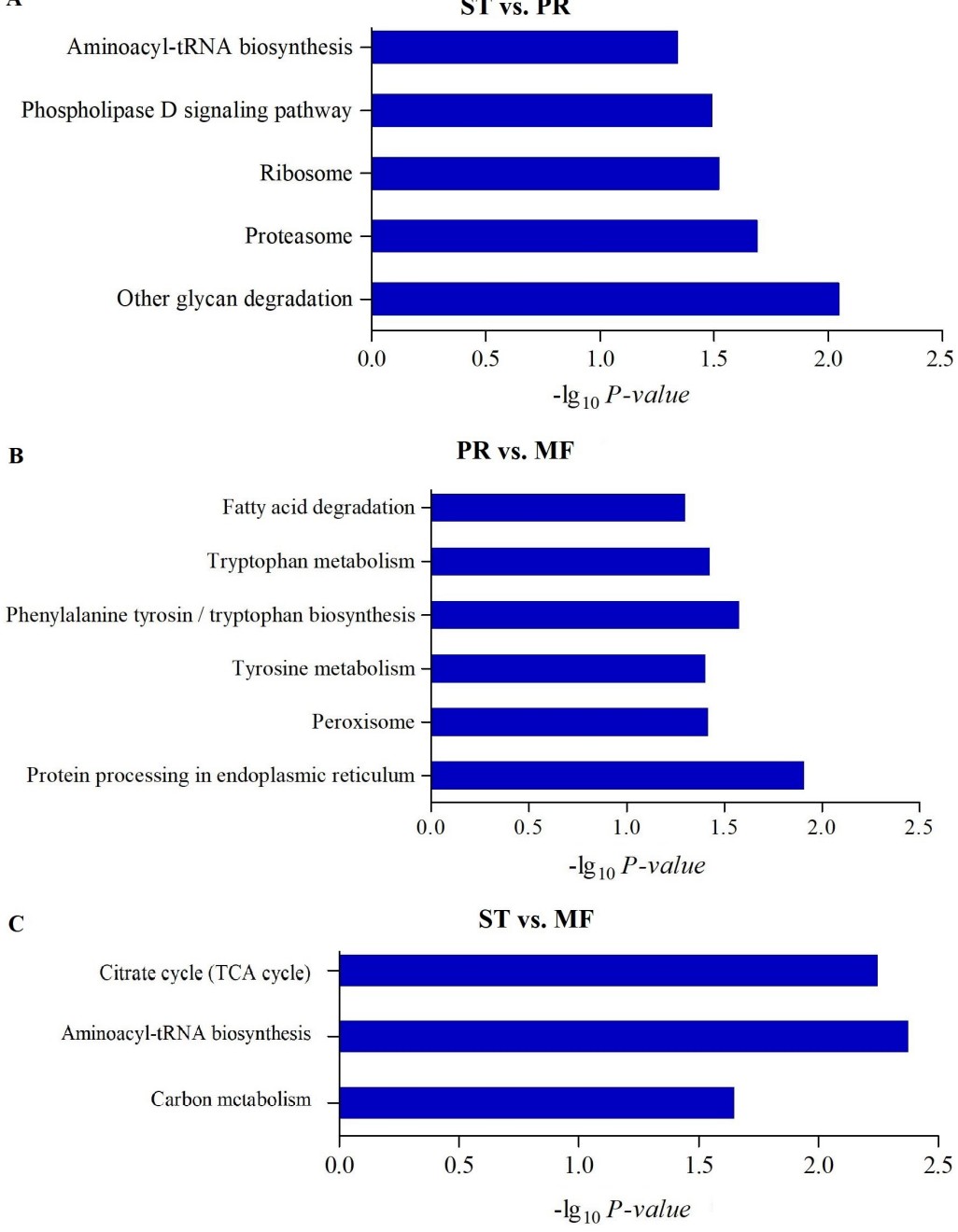

**Figure 4** **The enriched KEGG pathways of DAPs between different growth stages in *O. sinensis*.** The most enriched KEGG pathways of DAPs in the ST vs. PR comparison (A), in the PR vs. MF comparison (B) and in the ST vs. MF comparison (C). X-axes represent the enrichment score (−Log10 *P-value*)) of top GO terms enriched among DAPs. ST, sclerotium (mummified larva) before stroma development; PR, sclerotium with initial stroma (stroma < 1 cm); MF, fruiting body with mature perithecia, ascus and ascospores.

*hisB* leads to loss of cleistothecia production on medium with low levels of tryptophan or histidine, respectively (*Eckert et al., 1999*; *Eckert et al., 2000*; *Busch et al., 2001*). Moreover, when the amounts of a single amino acid is very low, both genes would activate varieties of amino acid biosynthesis (*Pöggeler, Nowrousian & Kück, 2006*), indicating that amino acid / protein synthesis would be involved in the fruiting body maturation. Besides, the pathway of peroxisome was up-regulated in MF compared to PR. Peroxisomes were detected to be required for the fruiting body formation and germination of sexual spore in *P. anserine* (*Eckert et al., 1999*). Here, long-chain acyl-CoA synthetase & acyl-protein synthetase (ACSLs, T5AQC4) involved in peroxisome was detected to be highly up-regulated in MF compared to PR. ACSLs is an essential enzyme for the synthesis of fatty acyl-CoA, which is mainly for synthesis of fatty acid and sterols (*Roche et al., 2013*). CAR1 is necessary for peroxisome biogenesis, and the *car1* mutant resulted in a sterile (*Peraza-Reyes & Berteaux-Lecellier, 2013*; *Berteaux-Lecellier et al., 1995*). Meanwhile, in peroxisome, mobilization of fatty acids by $\beta$-oxidation takes place, therefore, the sterile of *car1* mutant might be due to a disturbed fatty acid metabolism (*Pöggeler, Nowrousian & Kück, 2006*), suggesting that composition of fatty acids and their derivatives would be required for sexual development in *O. sinensis*. This is consistent with a high demand for energy during the fruit-body formation. On the other hand, peroxidases, SOD (T5A6F1, T5AJH5) and CAT (T5A5N4, T5AJ96), were detected up-regulated in MF compared to PR (Table S6). These enzymes convert ROS to harmless particles, protecting organism from oxidative damage by excessive ROS. Not only that, *sod-1* is necessary for correct fruiting-body morphology (*Pöggeler, Nowrousian & Kück, 2006*), and catalase-peroxidase gene *cpe* was detected in cleistothecia (*Scherer et al., 2002*), probably due to that *cpeA* act in concert to generate the correct amount of self-induced oxidative stress during the fruiting body formation (*Pöggeler, Nowrousian & Kück, 2006*), suggesting that ROS gradient may play roles in the development in *O. sinensis*. In ST vs. MF, the DAPs were enriched in the pathways of 'aminoacyl-tRNAs'(map00970), 'citrate cycle (TCA)'(map00020) and 'carbon metabolism (map01200)'. These pathways were detected to be up-regulated in ST compared to MF (Fig. 4C, Table S5). Some proteins (Table S6), that are involved in the latter two metabolisms, were identified to have a higher expression levels at the stage of ST, such as succinate-CoA ligase (ADP-forming) $\alpha$ subunit (T5A666), isocitrate dehydrogenase subunit (T5A5Y2) and pyruvate dehydrogenase $\alpha$ subunit (T4ZWD4). So high amounts of energy and carbohydrates might be needed for the pathogen-host interactions and hyphae growth at the stage of ST, as well as prepared for the latter fruiting process.

Additionally, fruiting body usually couldn't happen till severe stressors occur (*Holliday & Cleaver, 2008*). As far as we know, *O. sinensis* inhabits over 4,000 m above sea level in Qinghai-Tibetan Plateau. The fruiting-body induction and development in *O. sinensis* are tightly associated with the ecological factors specific to highland. Under certain environmental stresses, dikaryotic mycelia aggregate to form primordium, which marks the beginning of fruit body development (*Pöggeler, Nowrousian & Kück, 2006*). Here, heat shock proteins Hsp 70 (T5A2W7, T5ADA9, Table S6) were detected to be up-regulated in PR compared to ST. Heat shock treatment accelerated the fruiting body formation and sporulation of *Myxococcus xanthus* (*Otani et al., 2001*), suggesting that Hsp would be

induced by heat treatment and perhaps involved in the fruiting body formation. Besides, Hsp70 (T5ADA9, Table S6) has a higher expression in MF than that in ST, suggesting that this protein might be also important for fruit-body maturation. On the other hand, Hsps are also immunodominant antigens and major targets of host immune response during different types of infection (*Polla, 1991*). Our data showed that Hsp 30 (T5AGM5) and Hsp DanJ (T5AKH0) were up-regulated in ST compared to PR (Table S6). Hsp OS (T5A0N1) and Hsp 90 (T5AEC8) have a higher expression in ST compared to MF (Table S6). These results indicated that the host larva may produce Hsps for protection when infected by the pathogenic fungus. However, the mechanism needs further studies.

## Signal pathway in fruiting body formation of *O. sinensis*

Previous studies showed that mitogen-activated protein kinase (MAPK) pathway, regulate gene expression in response to extracellular stimuli that finally leads to fruiting body formation in *O. sinensis*. Our proteome data showed that the DAPs involved in MAPK signaling pathway, including G $\beta$ subunit , T5A9X0, KH domain RNA-binding protein, T5AF08, protein kinase like proteinT5A7J6 and two Ras proteins, T5AKF5, T5AA69, have a remarkably higher expression level in ST than that in the stage of MF (Table S6). In *A. nidulans*, *rasA* gene mutant cause aberrations in conidial germination and asexual development (*Fillinger et al., 2002*). In *N. crassa*, as well as other filamentous ascomycetes, it was demonstrated that subunits of G proteins are important for hyphal growth, conidiation, and fruiting-body development (*Yang, Poole & Borkovich, 2002*). Targeted disruption of two G $\alpha$ subunit genes in *Cryphonectria parasitica* revealed roles for the gene in fungal reproduction, virulence, and vegetative growth (*Pöggeler, Nowrousian & Kück, 2006*). Here, G $\beta$ subunit was detected to be up-regulated in ST compared to MF. RAS like protein subunits can regulate downstream effectors such as adenylyl cyclase (AC) and mitogen-activated protein kinase (MAPK) cascades. Besides, cAMP-dependent protein kinase regulatory subunit (T5ANJ5) was found to be up-regulated in ST. It was suggested that G-protein signaling would be involved in the infection and hyphal growth of *O. sinensis*. On the other hand, serine/threonine-protein phosphatase (T5AGS5, Table S6) involved in MAPK signaling pathway has a significantly higher expression level in MF compared to ST. Three different MAPKs and two different MAPKKKs have been proved to be involved in fruit-body development in different mycelial ascomycetes (*Pöggeler, Nowrousian & Kück, 2006*). Moreover, mutation of the corresponding genes always leads to multiple phenotypic defects in ascocarp formation. The MAPKKK NRC-1 of *N. crassa* is essential for female fertility (*Kothe & Free, 1998*). So MAPK signaling pathway may also be involved in fruit-body development in response to extracellular stimuli in *O. sinensis*.

## Hierarchical clustering analysis of DAPs involved in ROS metabolism

ROS play roles in the growth, development, defense responses against various stresses and that spatial regulation of ROS production is an important factor controlling the growth in fungi and plant. ROS production and scavenging is a dynamic oxidation–reduction process. Here, we identified the DAPs involved in ROS metabolism by looking for proteins predicted to be involved in oxidation–reduction processes. In ST vs. PR, ROS reflected in the GO

annotations includes GO:0051920 ,GO:0016491, GO:0016709, GO:0016717, GO:0055114 and GO:0003824 (Fig. 3A, Table S4) In PR vs. MF, ROS reflected in GO annotations that was related to ROS include GO:0006979, GO:0034599, GO:0072593, GO:0042743, GO:0042744, GO:0016209, GO:0016701, GO:0016697, etc. (Fig. 3B, Table S4). In ST vs. MF, ROS reflected in GO annotations include GO:0016491, GO:0004601 and GO:0016684. (Fig. 3C, Table S4). Among them, a total of 34 DAPs involved in ROS metabolism were identified (Table S2).

Furthermore, the protein profiles of DAPs involved in ROS metabolism across the three stages were clustered (Fig. 5, Table 2). The results demonstrated that there were four proteins clusters with visible difference expression patterns (Figs. 5A–5D). PR and MF stages were grouped together into one cluster indicating that PR and MF stages have more similar ROS related expression patterns than ST stage (Fig. 5E). In cluster 1, with 5 DAPs, had a gradual increase over the entire process from ST to MF, illustrating a different role for these proteins with respect to development, especially for maturation, including pyridine nucleotide-disulfide oxidoreductase (T5AE75), Superoxide dismutase (SOD, T5AJH5), electron transport protein (T5AHD1) and aldehyde dehydrogenase domain protein (T5AA56) (Table 2). For examples, in *N. crassa*, SOD-1 is necessary for light-dependent positioning of perithecial necks, probably due to generate a light-dependent ROS gradient that would control neck positioning (*Yoshida & Hasunuma, 2004*). SOD was successively up-regulated from PR to MF in *O. sinensis*, indicating that it may be play roles in the process of the entire fruiting body development, especially for the fruiting body maturation. As far as we know, light is required for the maturity by forming the sexual structures: sporulating structures that produce ascospore (*Pöggeler, Nowrousian & Kück, 2006*; *Degli Innocenti & Russo, 1983*), suggesting that SOD may be light-dependent regulation of sexual development in *O. sinensis*. Besides, electron transport proteins were located within the inner membrane of mitochondria, which is responsible for energy generation. Mutants of these proteins result in lack of energy and are sterile in *N. crassa* (*Duarte & Videira, 2010*). Electron transport protein (T5AHD1) were detected to gradually increase from ST to MF, indicating an increased demand of energy during the fruiting body development, especially for sexual development. Aldehyde dehydrogenase are putative indole receptor proteins involved in multicellular development and was found to be essential for fruiting body formation in *Stigmatella aurantiaca* (*Stamm, Lottspeich & Plaga, 2005*). Acetaldehyde dehydrogenase of the glyoxylate pathway can be induced by heat shock in *Myxococcus Xanthus* (*Yoon et al., 2002*) and was proposed to be relevant with fruiting body formation in *Flammulina velutipes* (*Otani et al., 2001*). This protein (T5AA56) was at the highest level in MF, indicating that it might be also required for the fruiting body maturation in *O. sinensis*, probably in response to temperature-related stress.

In cluster 2, with 6 DAPs, had a sudden increase in expression upon transition from ST to PR and then a sudden decrease in expression upon shifting from PR to MF phase, indicating that they may play a major role in the fruiting body initiation, such as histidine phosphotransferase (HPt, T5A92Z), one cytochrome P450 protein (T5A735) and Mannitol-1-phosphate 5-dehydrogenase (MPD, T5AIP9) (Table 2). For examples, MPD, a main enzyme for mannitol biosynthesis, was found to be abundance in the fruiting

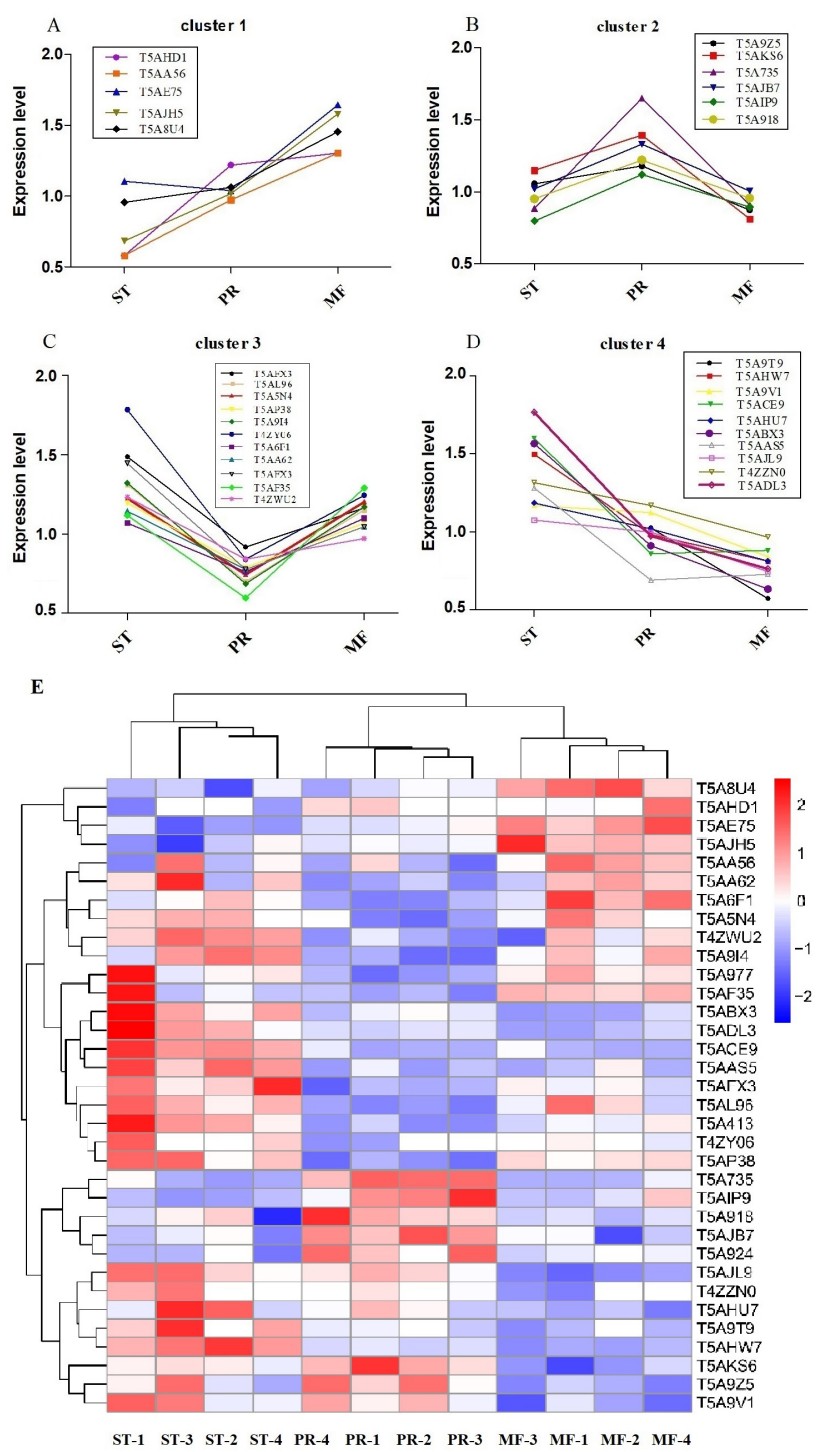

**Figure 5** **Clustering of reactive oxygen system (ROS) related DAPs expression profiles across the three growth stages in *O. sinensis*.** (A–D) Four protein clusters with different expression patterns. Overlaying curves of all ROS related DAPs within the cluster were shown. (E) The heat map reveals relative abundance of these ROS-related DAPs in different growth stages. ST, sclerotium (mummified larva) before stroma development; PR, sclerotium with initial stroma (stroma <1 cm); MF, fruiting body with mature perithecia, ascus and ascospores.

**Table 2  Hierarchical clustering analysis of oxidation-reduction related DAPs among the all comparisons.**

| Protein ID | ST vs. PR | p-value | PR vs. MF | p-value | ST vs. MF | p-value | Annotation |
|---|---|---|---|---|---|---|---|
| Cluster 1 | | | | | | | |
| T5AHD1 | 0.47 | 0.008 | 0.94 | 0.797 | 0.45 | 0.131 | Electron transport protein |
| T5AA56 | 1.13 | 0.433 | 0.75 | 0.026 | 0.85 | 0.224 | Aldehyde dehydrogenase domain protein |
| T5AE75 | 0.66 | 0.050 | 0.63146 | 0.0033 | 0.42 | 0.001 | Pyridine nucleotide-disulfide oxidoreductase |
| T5AJH5 | 0.68 | 0.158 | 0.64456 | 0.021 | 0.44 | 0.016 | Superoxide dismutase |
| T5A8U4 | 0.90 | 0.345 | 0.73109 | 0.006 | 0.66 | 0.006 | Bli-3 |
| Cluster 2 | | | | | | | |
| T5A9Z5 | 0.89 | 0.2805801 | 1.35 | 0.005 | 1.21 | 0.092 | Acetohydroxy acid isomeroreductase |
| T5AKS6 | 0.82 | 0.067688 | 1.72 | 0.005 | 1.42 | 0.009 | Formate dehydrogenase |
| T5A735 | 0.54 | 0.0009816 | 1.82 | 0.0004 | 0.97 | 0.841 | Cytochrome P450 |
| T5AJB7 | 0.77 | 0.0048387 | 1.33 | 0.012 | 1.02 | 0.883 | Isoflavone reductase family protein |
| T5AIP9 | 0.71 | 0.0057291 | 1.25 | 0.042 | 0.89 | 0.096 | Mannitol-1-phosphate 5-dehydrogenase |
| T5A918 | 0.78 | 0.0895424 | 1.27 | 0.015 | 0.99 | 0.961 | Protein disulfide-isomerase tigA |
| Cluster 3 | | | | | | | |
| T5AFX3 | 1.62 | 0.006 | 0.79 | 0.013 | 1.28 | 0.050 | Nor-1 |
| T5AL96 | 1.87 | 0.0008 | 0.61 | 0.016 | 1.14 | 0.364 | Catalase |
| T5A5N4 | 1.65 | 0.0008 | 0.62 | 0.017 | 1.02 | 0.856 | Catalase |
| T5AP38 | 1.52 | 0.002 | 0.74 | 0.0001 | 1.12 | 0.135 | Glucose-methanol-choline oxidoreductase |
| T5A9I4 | 1.93 | 0.004 | 0.59 | 0.003 | 1.13 | 0.356 | Alcohol dehydrogenase superfamily, zinc-type |
| T4ZY06 | 1.27 | 0.003 | 0.67 | 0.007 | 0.86 | 0.156 | Extradiol ring-cleavage dioxygenase |
| T5A6F1 | 1.50 | 0.054 | 0.69 | 0.011 | 1.04 | 0.807 | Superoxide dismutase |
| T5AA62 | 1.86 | 0.005 | 0.74 | 0.007 | 1.39 | 0.037 | Alcohol dehydrogenase superfamily, zinc-type |
| T5A413 | 1.87 | 0.165 | 0.46 | 0.0001 | 0.87 | 0.614 | L-threonine 3-dehydrogenase |
| T5AF35 | 1.47 | 0.001 | 0.86 | 0.304 | 1.27 | 0.065 | Transaldolase |
| T4ZWU2 | 1.27 | 0.003 | 0.68 | 0.007 | 0.861 | 0.156 | Mitochondrial peroxiredoxin PRX1 |
| Cluster 4 | | | | | | | |
| T5A9T9 | 2.10 | 0.042 | 1.79 | 0.048 | 3.77 | 0.015 | AhpC/TSA family protein |
| T5AHW7 | 1.52 | 0.0009 | 1.21 | 0.002 | 1.84 | 0.0002 | Flavin-containing monooxygenase |
| T5A9V1 | 1.04 | 0.659 | 1.34 | 0.011 | 1.39 | 0.021 | Glutathione peroxidase |
| T5ACE9 | 1.86 | 0.0007 | 0.98 | 0.836 | 1.81416 | 0.001 | WSC domain containing protein |
| T5AHU7 | 1.16 | 0.338 | 1.25 | 0.029 | 1.46 | 0.053 | NADPH–cytochrome P450 reductase |
| T5ABX3 | 1.71 | 0.043 | 1.44 | 0.046 | 2.46672 | 0.011 | NAD(P)-binding domain protein |
| T5AJL9 | 1.08 | 0.279 | 1.34 | 0.0003 | 1.44362 | 0.002 | 2-nitropropane dioxygenase |
| T5AAS5 | 1.85 | 0.002 | 0.95 | 0.718 | 1.75728 | 0.003 | Peroxiredoxin |
| T4ZZN0 | 1.12 | 0.124 | 1.21 | 0.026 | 1.36058 | 0.023 | Scavenger mRNA decapping enzyme |
| T5ADL3 | 1.82 | 0.038 | 1.27 | 0.037 | 2.31 | 0.017 | C-1-tetrahydrofolate synthase |

**Notes.**

ST, The mummified larvae coated with mycelia before stroma development;  PR, The samples of stroma with lengths <1 cm;  MF,  the fruiting body with mature ascus and ascospores.

body of *O. sinensis*, compared to that in mycelia  (*Feng et al., 2017*). As a result, it may increase the content of mannitol in *O. sinensis*, which was found to be related to fruiting body initiation and development of *A. bisporus* (*Kulkarni, 1990*). The mannitol content in the fruiting body of *A. bisporus* is about 8 to 20 times higher than that in mycelium (*Wannet*

*et al., 2000*). Our data showed that MPD was up-regulated in PR compared to the other two stages, suggesting that it may be related to initiation of the fruiting body. In our study, one cytochrome P450 protein was detected to be at the highest level in PR. These enzymes are involved in secondary metabolism, oxidative and peroxidative. The mutant of eln2 gene, encoding a novel type of cytochrome P450, resulted in the mutant phenotype of primordial shaft (*Muraguchi & Kamada, 2000*). One explanation is that a changed catalytic activity may generate a toxic compound that affects development in the primordial shaft (*Muraguchi & Kamada, 2000*). The relationship between cytochrome P450 and the fruiting body initiation is the focus of another project in our lab. Besides, HPt functions as ROS sensors and up-regulates stress-activated MAP kinase cascade, which was shown to be involved in fruiting body development in different mycelial ascomycete (*Fassler & West, 2013*). This protein (T5A924) was up-regulated in PR, suggesting that the initiation of the fruiting body may be depend on MAPK pathways.

Cluster 3, with 11 DAPs, suddenly decreased from the ST to PR stage and then suddenly increased from the PR to the MF stage, including catalase (CAT, T5AL96, T5A5N4), superoxide dismutase (SOD, T5A6F1), transaldolase (T5AF35), Nor-1 (T5AFX3) and glucose-methanol-choline oxidoreductase (GMCA, T5AP38), etc. (Table 2), indicating that these proteins may play roles in the growth of ST, as well as the maturation of the fruiting body. For examples, ROS generated by microbial NADPH oxidase (NoxA) would regulate sexual development in *A. nidulans* (*Lara-Ortíz, Riveros-Rosas & Aguirre, 2003*). NOX2, together with NOR1, controls ascospore germination in *Sordaria macrospora* (*Dirschnabel et al., 2014*). Two NOX isoforms are required for sexual reproduction and ascospore germination in *P. anserine* (*Malagnac et al., 2004*). Our data showed that Nor-1 was up-regulated in stages of PR and MF compared to ST, suggesting that NOX1-Nor complex system may also contribute to generate ROS, triggering spore germination in *O. sinensis*. This is the very important regulatory pathway of ROS generation and may be an integral part of fruiting-body formation in fungi. Besides, the mutant of GMCA in *A. nidulans* resulted in the suppression of asexual development. GMCA was also the target of FlbB, a upstream developmental activator, inducing *A. nidulans* asexual differentiation (*Etxebeste et al., 2017*), thereby, GMCA may also play a role in the growth of hyphae and morphogenesis of sclerotia in *O. sinensis*. Besides, transaldolase is important for the balance of metabolites in the pentose-phosphate pathway (PPP), which is the major pathway for glucose metabolism, the https://www.sciencedirect.com/topics/chemistry/tricarboxylic-acid cycle (TCA) and carbohydrate catabolism (*Qian et al., 2008*). Our data showed transaldolase is expressed at a much higher level in ST and MF than that in PR, suggesting that higher amounts of energy may be in the form of carbohydrate initially stored in the hyphae of ST and required for the fruiting body development.

In cluster 4, with 10 DAPs, has a gradual decrease across the three stages and at the highest expression level in ST, including NADPH–cytochrome P450 reductase (T5AHU7), WSC domain-containing proteins (T5ACE9), 2-nitropropane dioxygnease (2-NPD, T5AJL9), Glutathione peroxidase (T5A9V1) and peroxiredoxin (T5AAS5), etc. (Table 2). For examples, the deletion mutants of five WSC domain-containing proteins showed a delay of germination and a decrease of conidial UV-B resistance, thermotolerance or both and

indicated that a significance of each WSC protein for the *Beauveria bassiana* adaptation to host insects (*Tong et al., 2016*). This protein has a higher level in ST and PR compared to MF in *O. sinensis*, suggesting that it may contribute to *O. sinensis* adaptation to host insect or harsh physical stresses. 2-NPD, a flavin dependent enzyme, catalyzes the oxidation of nitronates to their corresponding carbonyl compounds and nitrite (*Gadda, Francis & Belaineh, 2007*). 2-NPD was demonstrated to be involved in vegetative growth in *Pyronema confluens* (*Minou & Kück, 2010*). Our data showed that this protein was at the highest level in ST, suggesting that it may be linked with the vegetative growth or asexual development in *O. sinensis*. The *O. sinensis* genome displayed a considerable expansion of gene families that are mainly involved in fungal pathogen, peroxidase included (*Xia et al., 2017*). Peroxiredoxin is regarded as one of the most prominent and integral component in responding to different levels of oxidative stress in *Vibrio vulnifcus* (*Bang, Oh & Choi, 2012*; *Alharbi et al., 2019*). So these proteins may functions as signal transducers in various stress responses in *O. sinensis*. Besides, NADPH–cytochrome P450 reductase may be associated to energy production and conversion for the whole developmental process in *O. sinensis*.

ROS is critical for sexual fruiting body development in filamentous fungi. In this study, the proteins involved in ROS metabolism system were proposed to play diverse roles in different growth stages in *O. sinensis*. They may play a role in fungal pathogenesis at the stage of ST, be related to the induction of differentiation processes in PR and trigger sexual reproduction in MF. So their exact roles and the underlying mechanism are likely to be stage-specific, which will be the focus of our another project.

## qRT-PCR Analysis

To confirm the reliability of the iTRAQ sequencing data, six proteins with different direction of accumulation were selected for qRT-PCR validation, included SOD (T5A6F1) , CAT (T5A5N4), NOR-1 (NADPH oxidases receptor-1, T5AFX3), NDK (nucleotide diphosphate kinase, T5ALL5), CYT(cytochrome, T5AKK7) and His2A (Histone 2A, T5A212). The gene-specific primers used in qRT-PCR are listed in Table S7. According to the results of qPCR, in the comparison of ST vs. PR, except for one protein (T5A6F1) encoding gene, the expression changes of five genes detected by qPCR were similar to the direction of fold change acquired by the iTRAQ sequencing results, including four genes up- regulated, one gene down-regulated in ST compared to PR and one gene similarly expressed between the two stages (Fig. 6A). In PR vs. MF comparison, except for two proteins (T5AFX3, T5AKK7) encoding genes , the qPCR results of four genes were similar to the results of proteome analysis, including three genes up-regulated in PR, two genes down-regulated in PR and one genes similarly expressed between the two stages (Fig. 6B). So our transcription patterns of these proteins support the results of the proteomic analysis.

## ROS accumulation, SOD and CAT activities assay

In order to explore the effect of ROS on the fruiting body development in *O. sinensis*, we analyzed the accumulation ROS in the crucial growth stages (ST, PR and MF), respectively. The ROS accumulation was reflected by a fluorescence intensity per g tissue. MF showed about ROS level significantly increased by about 1.79- and 1.58-fold in MF, compared to

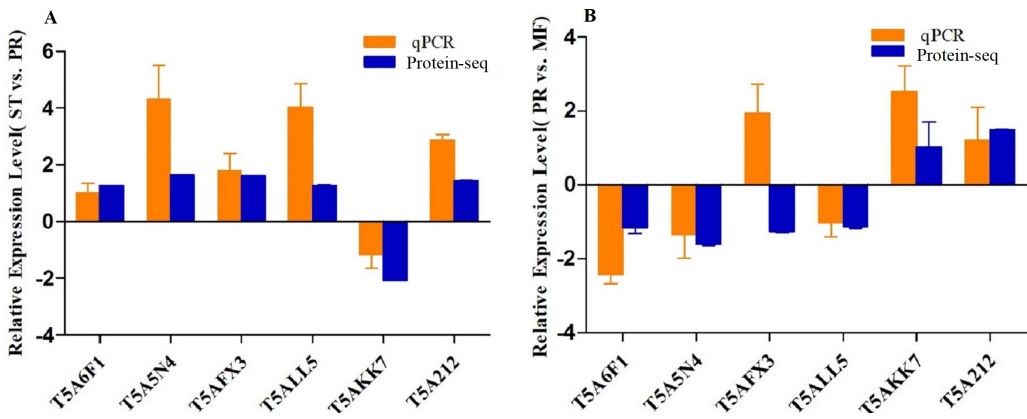

**Figure 6   qRT-PCR validation of the genes encoding for the DAPs.** (A) Bars represent the fold change in expression of each candidate gene identified in ST relative to PR. (B) Bars represent the fold change in expression of each candidate gene identified in PR relative to MF. Orange bars represent qRT-PCR result ($2^{\Delta\Delta Ct}$). Error bars indicate the standard error. Blue bars represent the protein-seq results . 18sRNA was the internal reference. ST: sclerotium (mummified larva) before stroma development; PR: sclerotium with initial stroma (stroma <1 cm); MF: fruiting body with mature perithecia, ascus and ascospores.

ST and PR, respectively (both $p < 0.001$), while there was no significant difference between ST and PR (Fig. 7), suggesting that ROS may be positively relevant to the fruiting body maturation in *O. sinensis*. In order to investigate the site of ROS production within hyphae and fruiting body, sections of the hyphae from ST and the fruiting body part from MF were stained with DCFH-DA, respectively. The result showed that ROS signals remarkably distributed within the conidiospore and ascus (Fig. 8), suggesting that ROS may promote the growth of hyphae and ascus formation. Previous study showed that Nox-generated ROS controlled sexual development in fungi (*Pöggeler, Nowrousian & Kück, 2006*), similar to our study.

Environmental stresses, such as intensive UV, low temperature and fungal-insect interaction, usually result in excessive ROS accumulation in *O. sinensis* (*Xia et al., 2017*). Our proteome data showed that the DAPs were enriched in GO term of 'cellular response to oxidative stress' in ST vs. MF. It is consistent with an increased ROS level in MF. Our data showed SOD activity remarkably increased about 3.81-fold in PR compared to ST (*p-value* <0.01), about 1.39-fold in MF compared to PR (*p-value* <0.05) and about 5.3-fold in MF compared to ST (*p-value* <0.001) (Fig. 9A), similar to our proteome data (Table S1). SODs catalyze dismutation of superoxide radical to produce $H_2O_2$ and $O_2$ and is considered to be the first line of defense against oxidative stress in eukaryotic cells. SOD has a significant and serial increase across the three stages, indicating that it may be produced to maintain cellular redox homeostasis and prevent the cellular damage. Upon increased gene expression of Cu, Zn-SOD and Mn-SOD, the lifespan and resistance to oxidative stress also increase in the cells of *S. cerevisiae* (*Gessler, Aver'yanov & Belozerskaya, 2007*). On the other hand, SOD mutants of *N. crassa* were distinguished by reduction of sexual reproduction and decreased ability for formation of conidia (*Gessler, Aver'yanov & Belozerskaya, 2007*). SOD-1 involved in generating a light-depended ROS gradient controls neck positioning in

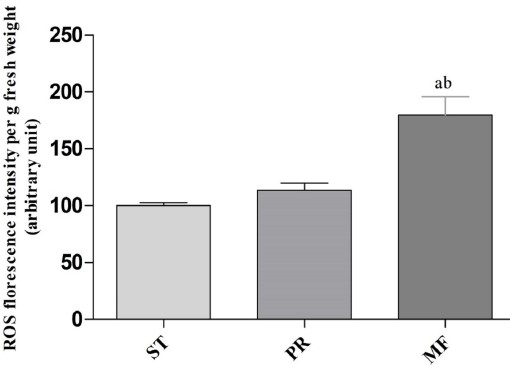

**Figure 7** **ROS accumulation in samples of different growth stages of *O. sinensis*.** ST, sclerotium (mummified larva) before stroma development; PR, sclerotium with initial stroma (stroma <1 cm); MF, fruiting body with mature perithecia, ascus and ascospores. Error bars indicate the standard error. Different letters indicate statistically difference between the comparisons, (a) *p* < 0.001 (MF relative to ST); (b) *p* < 0.001 (MF relative to PR).

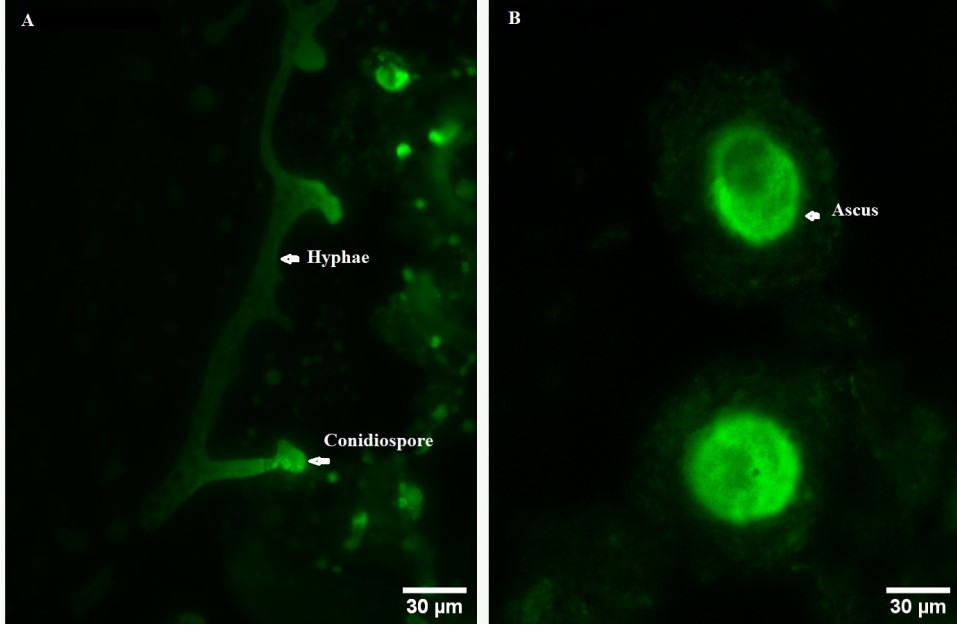

**Figure 8** **ROS measurement via DCFH-DA probe detection in *O. sinensis*.** (A) Hypha was detected by 5 μM H2DCF-DA for 20min at 37 °C. (B) Fruiting body was detected by 5 μM H2DCF-DA for 20min at 37 °C. Scale bar represents 30 μm.

*N. crass* (*Pöggeler, Nowrousian & Kück, 2006*), suggesting similar mechanism in *O. sinensis*. Besides, CAT activity significantly increased by about 3.95- and 3.73-fold in MF, compared to ST and PR (both *p-value* <0.001), respectively, while there was no significant difference between ST and PR (Fig. 9B). In *A. nidulans, nox* A is induced during sexual development, at this time, catalase-peroxidase gene is transcriptionally, converting the ROS to harmless

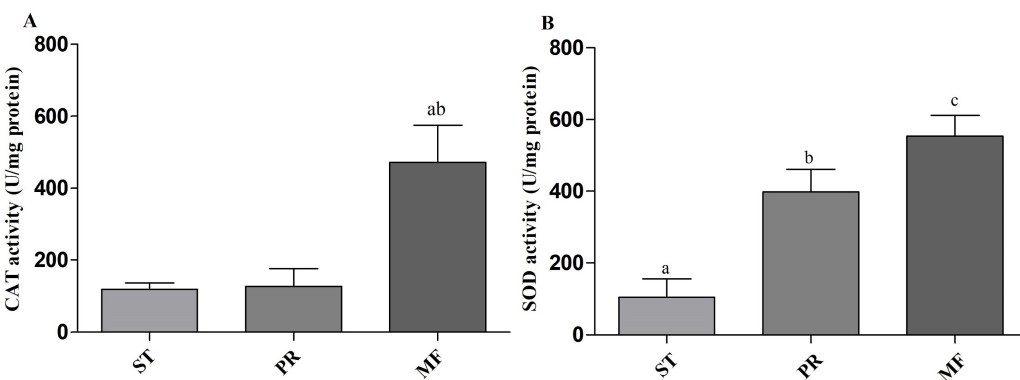

**Figure 9** **CAT and SOD activities in samples of different stages in *O. sinensis*.** (A) CAT activity in samples at different stages. (a) $p < 0.01$, MF relative to ST; (b) $p < 0.01$, MF relative to PR. (B) SOD activity in samples at different stages. (a) $p < 0.01$, PR relative to ST; (b) $p < 0.001$, MF relative to ST; (a) $p < 0.05$, MF relative to PR. ST, sclerotium (mummified larva) before stroma development; PR, sclerotium with initial stroma (stroma < 1 cm); MF, fruiting body with mature perithecia, ascus and ascospores. Error bars indicate the standard error. Different letters indicate statistically difference between the comparisons.

compounds, probably protecting cell from oxidative stress (*Pöggeler, Nowrousian & Kück, 2006*). In *N. crassa*, CAT-1 activity in spores was found to be 60 times higher than that in the mycelium (*Diaz et al., 2005*). The generation and degradation of ROS level was disturbed, suggesting that ROS may act in gradient-dependent manner to regulate the fruiting body development of *O. sinensis*. The underlying mechanism needs further studies.

## CONCLUSION

In this study, we used an iTRAQ quantitative approach coupled with LC-MS/MS and bioinformatics to investigate the proteomic basis of fruiting body development. Samples of three major stages (ST, PR and MF) were collected to be examined. A total of 1,875 DAPs were identified. GO enrichment and KEGG analysis demonstrated that 'primary metabolisms', 'response to oxidative stress', 'antioxidative activity' were enriched during the fruiting process. Furthermore, some crucial development-related proteins were identified, such as Hsps, serine / threonine-protein phosphatase and serine proteases. The protein profiles of the DAPs involved in ROS system of the three crucial stages were identified, suggesting that ROS system may play stage-specific roles in different stages through integrating with the baseline metabolism pathways. Besides, a stronger ROS accumulation was detected at the stage of MF compared to the other two stages. Observed under a fluorescence inverted microscopy, ROS signals distributed within the conidiospore and ascus. SOD activities serially increased across the three stages and CAT significantly increased in MF compared to the other two stages. Combined with previous studies, we proposed that ROS gradient may control the development guided by environmental stimulus. The finding would help to further understand the biology and impact the big industry of this valuable fungus.

## Funding

This study was supported by the Natural Sciences Foundation of China Science (81872959, 81373920, 30801522), Yong Science and Technology Innovation Team of Sichuan Province (2019CXTD0055), China Scholarship Foundation (201708510027), and the Xingling Scholar Discipline Talent Research Promotion Plan (ZYTS2019017). The funders had no role in study design, data collection and analysis, decision to publish, or preparation of the manuscript.

## Grant Disclosures

The following grant information was disclosed by the authors:
Natural Sciences Foundation of China Science: 81872959, 81373920, 30801522.
Yong Science and Technology Innovation Team of Sichuan Province: 2019CXTD0055.
China Scholarship Foundation: 201708510027.
Xingling Scholar Discipline Talent Research Promotion Plan: ZYTS2019017.

## Competing Interests

The authors declare there are no competing interests.

## Author Contributions

- Xinxin Tong conceived and designed the experiments, performed the experiments, analyzed the data, prepared figures and/or tables, authored or reviewed drafts of the paper, and approved the final draft.
- Fang Wang and Xinyi Jiang performed the experiments, authored or reviewed drafts of the paper, and approved the final draft.
- Han Zhang analyzed the data, authored or reviewed drafts of the paper, and approved the final draft.
- Jing Bai and Pan Yue performed the experiments, prepared figures and/or tables, and approved the final draft.
- Qiang Dong analyzed the data, prepared figures and/or tables, and approved the final draft.
- Xinrui Li performed the experiments, prepared figures and/or tables, authored or reviewed drafts of the paper, and approved the final draft.
- Li Wang performed the experiments, authored or reviewed drafts of the paper, and approved the final draft.
- Jinlin Guo conceived and designed the experiments, authored or reviewed drafts of the paper, and approved the final draft.

## Data Availability

Data are available at Integrated Proteome Resources (iProX): PXD021260.
https://www.iprox.org//page/project.html?id=IPX0002428000.

## Supplemental Information

Supplemental information for this article can be found online at http://dx.doi.org/10.7717/peerj.10940#supplemental-information.

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
