# Peer review of "iTRAQ-based comparative proteome analyses of different growth stages revealing the regulatory role of reactive oxygen species in the fruiting body development of Ophiocordyceps sinensis"

_PeerJ, doi:10.7717/peerj.10940_

## Round 0.1 · original submission · Major Revisions

Your manuscript was reviewed by two independent reviewers. Both reviewers feel the manuscript is interesting but have also pointed out several problems. Please revise the manuscript based on the reviewers' comments.

Reviewer 1 ·

Basic reporting

This manuscript presents an iTRAQ based proteomic analysis of three developmental stages of the medicinal enthomopathogenic fungus Ophiocordyceps sinensis, focusing on the role of ROS in fruiting body development.
The English of the manuscript is understandable but definitely needs improvement in order to be correct and precise. Some remarks have been posted in General comments.
Background. The Introduction is concise and well-focused but in my opinion more information should be given about the biology and artificial cultivation of Cordyceps sinensis, as well as on omics research concerning this fungus.
The structure of the manuscript is adequate and conforms to PeerJ standards but some improvement in data presentation is necessary (see remarks on figures).
Raw data are supplied as supplementary files.
Literature used is relevant but some important sources are not included. For example, about biology of Cordyceps sinensis - Li, M., Meng, Q., Zhang, H., (...), Qin, Q., Zhang, J. Vegetative development and host immune interaction of Ophiocordyceps sinensis within the hemocoel of the ghost moth larva, Thitarodes xiaojinensis. 2020 Journal of Invertebrate Pathology 170,107331; Liu, G., Han, R., Cao, L., Schmidt-Jeffris, R. Artificial Cultivation of the Chinese Cordyceps from Injected Ghost Moth Larvae 2019 Environmental Entomology 48(5), pp. 1088-1094. For participation of ROS in fruity body development see also Liu, Q., Zhao, Z., Dong, H., Dong, C. Reactive oxygen species induce sclerotial formation in Morchella importuna 2018 Applied Microbiology and Biotechnology 102(18), pp. 7997-8009.

Experimental design

This is an Original primary research within Aims and Scope of the journal. The Research question is relevant and meaningful but could be better defined to emphasize how research fills this knowledge gap. The methods are not described with sufficient detail & information to replicate (see General comments).

Validity of the findings

no comment

Additional comments

In my opinion, the manuscript needs a major revision in order to be published.
Major remarks
Abstract
Line 17 – “its potential mechanisms” – unclear, the meaning is lost
Line 18 – the statement “protein patterns between PR and MF were more similar than ST” is in concert with the main text but it is not in concert with the next sentence about the numbers of DAPs. In the text, there are 510, 173 and 514 DAPs detected in the comparisons of ST vs. PR, PR vs. MF, and ST vs. MF, respectively.
Line 21 – “enriched in terms of…” did you mean enriched in proteins related to…
Line 22 – “enriched in pathways of..” more convenient is here enriched in terms…. (pathways of peroxidase does not sound well)
Line 25 – “R package” – not clear
Line 29 – I wonder about electron transport protein (T5AHD1) and histidine phosphotransferase (HPt, T5A9Z5) - are they ROS-related proteins? Especially histidine phosphotransferase is a component of signal transduction system – see Front. Chem., 19 May 2017 | https://doi.org/10.3389/fchem.2017.000 Fungal Histidine Phosphotransferase Plays a Crucial Role in Photomorphogenesis and Pathogenesis in Magnaporthe oryzae
The conclusion in the abstract : This findings firstly offer the perspective of proteome to understand the biology of fruiting body development and highland adaptation, which inform the future large-scale artificially breading of O. sinensis. – needs better elaboration.

The Introduction needs improvement. I find some contradiction in the statement “Till now, It has not yet commercially cultivated” (line 49) and the cited reference Li et al in 2019, Critical Reviews in Biotechnology 39(2), pp. 181-191 “ omics studies, including genomic, transcriptomic, proteomic, and metabolomic studies, have helped to understand the biology of the fungus, the success of the artificial cultivation of the Chinese cordyceps is clearly a milestone “
More info should be added in the first paragraph on the biology and medical importance of this fungus and the problems of natural exhaustion of these resources and difficulties in cultivation for commercial purposes, as well as the scarcity of data for understanding the biology of development of the fruiting body.
Recent omics research (genomic, transcriptomic, proteomic) on this species should be briefly stated. Some insights from transcriptomic studies (more references should be used) - refs not cited: Zhao, Y., Zhang, J., Meng, Q., (...), Chen, C., Qin, Q. Transcriptomic analysis of the orchestrated molecular mechanisms underlying fruiting body initiation in Chinese cordyceps. Gene 2020, 763, 145061. Proteome analyses up to date should be summarized - for ex. Zhang, X., Liu, Q., Zhou, W., (...), Qi, L.-W., Yin, X. A comparative proteomic characterization and nutritional assessment of naturally- and artificially-cultivated Cordyceps sinensis. 2018 Journal of Proteomics 181, pp. 24-35. Other proteomic studies - the cited as ref 32. Feng K , Wang L Y , Liao D J , et al. (2017). "Potential molecular mechanisms for fruiting body formation of Cordyceps illustrated in the case of Cordyceps sinensis. " Mycology, 8(4):1-28; the non-cited Dong, Y.-Z., Zhang, L.-J., Wu, Z.-M., (...), Ni, L., Zhu, J.-S. Altered proteomic polymorphisms in the caterpillar body and stroma of natural Cordyceps sinensis during maturation. 2014 PLoS ONE 9(10),e109083.
Lines 49-51 – the sentence “The understanding of biology of the fruiting body development not only helps to realize the large-scale industry of this fungus but also protect this precious resources for sustainable usage” should be rephrased, it is not very convincing in the above content. Something like: This comparative proteomic study fills the gap in knowledge and provides useful information for sustainable usage and possible cultivation of this fungus in a large scale industry.
Minor remarks in Introduction:
Line 50 – realize=become aware, other meanings of realize=implement, fulfill, materialize, accomplish, achieve. Please chose more appropriate synonym.
Line 54 – “produce the ROS antioxidant defense” – please replace “produce…” with “develop an adequate ROS defense”
Line 55 – “precious” or previous?
Line 57 - conditions have/ condition has
Line 63 – “genes involved in peroxidases” - please replace with “genes coding for peroxidases”
Lines 70,71 – sentences beginning with numbers (12, 18) – should be written in letters (Twelve, eighteen)
Line 70 – DEGs should be explained at first mentioning
Line 76 – based on the above evidence…
Line 77 – with – unclear. Do you mean along with? guided by?
Line 81 - proteome was/ proteomes were

MMs - several small points of incorrectness
Collection of O. sinensis samples – How was mycelia separated from the dead larvae? Could you identify some animal proteins in ST samples? Did you separate ascospores from the mycelium at MF stage?
line 91 – “Traditional Chinses Medicine” – Chinese?
Protein preparation – line 97 – cocktail of what? IAA – defined only at line 104 with abbreviation IAM; TEAB - Triethylammonium Bicarbonate – defined only at line 110, first mentioning line 100. All abbreviations should be defined at first mentioning. Line 101 “Bradford Protein Assay Kit according to the manufacturer’s instructions” - which was the manufacturer? Specify the way it was done for trypsin. Line 108 – peptides; line 114 – “on Q Extractive” or Q Exactive? ACN – acetonitrile? FA – formamide? Line 147-148 – unclear sentence, the second werb “was” – not necessary.
Lines 160 -167 Reactive oxygen species (ROS) measurement – missing reference.
Line 169 – principle of methods for CAT and SOD activities should be given in brief.
Line 176 – root samples?
How many biological repeats were included – three (lines 186,189) or four (lines 144-145, 198)?
Lines 158-159 – R package – unclear.

Results and discussion:
Line 196 “Quality proteomic analysis” – unclear. An overview?
Line 210 – biological replicates are explained in MMs, not necessary to repeat.
Line 215 – “significant proteins” – unclear. DAPs?
Line 216 –“.And” – do not begin sentence with an “and”
Line 230 – omit “serial”
Line 231 – “molecular function (MF)” but MF is abbreviation for mature fruiting body
Line 241 – “paly” or play?
Line 242 –especially in the response
Line 247 – omit “that”
Line 272 – “into activate varieties of amino acid” – unclear
Line 279 – “has defects in a sterile” – please reformulate
Line 284 – this sentence is highly speculative
Line 400 - ROS signaling pathway
Line 402 –403 - a reference is necessary.
Line 404 - were enriched in
Line 405-406 –“ SOD and CAT are important antioxidant enzymes to eliminate harmful free radicals – please add more details – SOD converts superoxide radicals to hydrogen peroxide whereas CAT detoxifies hydrogen peroxide – different functions. Depending on SOD/CAT ratio different forms of ROS will be more abundant
Line 410 – “SOD has an increase tendency on protein level “SOD has a tendency to increase at the protein level

Conclusion
Line 423 – “investigate the genetic basis” – unclear, this is a proteomic study
Lines 431-434 How the authors could explain “a stronger ROS accumulation was detected at the stage of MF compared to stages of ST and PR. Meanwhile, SOD activity significantly increased across the three stages and the activity of CAT significantly increased in MF compared to the other two stages”? – both ROS increased and their scavengers also increased at MF. ROS gradient should be confirmed histochemically with specific staining for ROS forms.

Figure 2B – “The number of DAPs is shown on the top of histograms” – not seen
Figure 3 - enriched GO functional classification of DAPs. The green bars represent cellular components; The orange represent biological processes; the blue bars represent molecular functions. – Usually cellular components, biological processes and molecular functions are given in separate charts, not mixed.
DEGs –please define.
Figure 6 – black bars, white bars - but bars are in colours.
Figure 7 – Y axis – the title should be more exact (not – ROS fluoresncence)
Table S5 - the enriched KEGG pathway in the comparisons – alcoholism or alcohol metabolism?

Reviewer 2 ·

Basic reporting

Authors are advised to proof read the manuscript to identify the grammatical errors including sentence construction and word usage.

Experimental design

no comment

Validity of the findings

The current study provides proteomic analysis of the developmental stages of O. sinensis with specific focus on the fruiting body as a follow up to the previous publication by the authors (Tong X et al; Peer J; 2020) which analyzed the transcriptomic differences in the same. The current study provides data to suggest involvement of ROS signal transduction in the development stages of the fungi. The authors provide sufficient literature to corroborate their data.

However, the authors are advised to assess the novelty of the study in greater detail and add the lacunae in the field to emphasize the need to undertake the study.

Major Comments:
• The aim of the paper; as stated by the authors is to discuss the underlying biology of the development of fungus towards the overarching goal of providing better understanding for eventual commercial cultivation. In view of this, it would be pertinent to also discuss the DAPs and associated pathways (if any) exclusive to each development stage.
• The authors provide data to implicate ROS and antioxidant (CAT/SOD) levels as major regulatory factors in development of O. sinensis. It would also be pertinent to explore and demonstrate experimentally whether perturbations in the two could actually result in developmental delay/abnormalities.

---

## Round 0.2 · Minor Revisions

Please answer comments from reviewer 1.

Reviewer 1 ·

Basic reporting

no comment

Experimental design

no comment

Validity of the findings

no comment

Additional comments

The revised ms has been substantially improved, however some small mistakes still remain.
Line 106 – “containing of urea lysis buffer containing 8M Urea” – repetition, plrase omit the first “containing of”
Line 107 – are you sure that “ultrasonic pyrolysis” is pyrolysis? Please check.
Line 112 – then samples were kept
Line 116 – to obtain urea concentrations
Line 130 – followed by what? Unclear
Line 135 – 136% solvent A – are you sure?
Line 247 – were oxidoreductases
Line 248 - functions in carbohydrate transport
Line 315 – harmless particles
References should be written according to the requirements of the journal. At some places, italics is not used for Latin names of species.
Figure 4 – ST vs PR – ordinate axis – as far as I know “Systemic lupus erythematosus” is an human disease

Reviewer 2 ·

Basic reporting

The revised manuscript now conforms to PeerJ standards of clear and unambiguous reporting with relevant literature cited and professional English.
The authors have acknowledged the comments and incorporated the data in the revised manuscript addressing the comparison of DAPs at each stage of the fruiting body development of O. sinensis and have provided tabulated data for clarity. They have also tried to assess the ROS generation experimentally to clarify the importance of SOD/CAT in the hyphae and fruiting body while acknowledging and clearly stating future studies to be undertaken for mechanistic advancement.

Experimental design

Additions to the material and method section now provide sufficient details to replicate the study.

Validity of the findings

The authors have now addressed the novelty of study in the introduction with sufficient literature citation.
The conclusions are accompanied by valid speculations and limitation of the study.

---

## Round 0.3 · accepted · Accept

I am satisfied with authors response and changes to the manuscript.